# Rapid Identification of Benign Gallbladder Diseases Using Serum Surface-Enhanced Raman Spectroscopy Combined with Multivariate Statistical Analysis

**DOI:** 10.3390/diagnostics13040619

**Published:** 2023-02-08

**Authors:** Wubulitalifu Dawuti, Jingrui Dou, Jintian Li, Hui Liu, Hui Zhao, Li Sun, Jin Chu, Renyong Lin, Guodong Lü

**Affiliations:** 1State Key Laboratory of Pathogenesis, Prevention, and Treatment of Central Asian High Incidence Diseases, Clinical Medical Research Institute, The First Affiliated Hospital of Xinjiang Medical University, Urumqi 830054, China; 2School of Public Health, Xinjiang Medical University, Urumqi 830054, China; 3Department of Clinical Laboratory, The First Affiliated Hospital of Xinjiang Medical University, Urumqi 830054, China

**Keywords:** surface-enhanced raman spectroscopy (SERS), gallbladder (GB) stone, gallbladder (GB) polyp, serum, orthogonal partial least squares discriminant analysis (OPLS-DA), diagnosis

## Abstract

In this study, we looked at the viability of utilizing serum to differentiate between gallbladder (GB) stones and GB polyps using Surface-enhanced Raman spectroscopy (SERS), which has the potential to be a quick and accurate means of diagnosing benign GB diseases. Rapid and label-free SERS was used to conduct the tests on 148 serum samples, which included those from 51 patients with GB stones, 25 patients with GB polyps and 72 healthy persons. We used an Ag colloid as a Raman spectrum enhancement substrate. In addition, we employed orthogonal partial least squares discriminant analysis (OPLS-DA) and principal component linear discriminant analysis (PCA-LDA) to compare and diagnose the serum SERS spectra of GB stones and GB polyps. The diagnostic results showed that the sensitivity, specificity, and area under curve (AUC) values of the GB stones and GB polyps based on OPLS-DA algorithm reached 90.2%, 97.2%, 0.995 and 92.0%, 100%, 0.995, respectively. This study demonstrated an accurate and rapid means of combining serum SERS spectra with OPLS-DA to identify GB stones and GB polyps.

## 1. Introduction

Benign gallbladder (GB) disease usually presents as luminal lesions and localized or diffuse thickening of the GB wall. GB stones and GB polyps are the two most prevalent benign disorders and have a 5–10% probability of becoming cancerous (malignant) [1]. These benign diseases include adenomyomatosis, acute cholecystitis and others [2], which exhibit a range of clinical signs and symptoms. Patients may be asymptomatic or may suffer from acute biliary colic, jaundice and fever. Required treatment and management strategies differ accordingly. In addition, benign GB diseases can resemble GB cancers and present with a variety of imaging appearances [3,4]. Therefore, differentiating these diseases for the purposes of therapy and prognosis is essential.

Currently, abdominal B ultrasound, computed tomography (CT) and other imaging technologies are commonly used to identify GB stones and GB polyps, but they all require expensive software and hardware as well as visual observation of imaging physicians to determine results. They also have the problem of low sensitivity [5,6,7]. Laboratory results have revealed the leukocytosis with a left shift and minimal increase in the levels of bilirubin and alkaline phosphatase. Overall, GB stones and GB polyps can be neither confirmed nor ruled out by a single clinical finding or laboratory test [3]. Therefore, finding a rapid and accurate diagnostic method to identify benign GB disease is necessary.

Surface-enhanced Raman spectroscopy (SERS) is a method of boosting the Raman signals of biomolecules by utilizing nanometerized metal substrates, including metal colloids [8]. SERS makes it easier to identify alterations in the molecular “fingerprint” of biological fluids related to cancer, including blood [9], urine [10] and saliva [11]. SERS has become common in the examination of biofluids for diagnosing various illnesses, particularly cancer [12,13]. Several studies have been conducted on SERS of blood serum or plasma samples [14,15,16,17]. Feng et al. [18] used silver sol as an active substrate for SERS in conjunction with plasma detection to provide a simple and noninvasive method for plasma detection of nasopharyngeal cancer. After analyzing the SERS spectra of 43 patients with nasopharyngeal cancer and those of 33 healthy individuals, the researchers discovered that when compared to the plasma of normal people, that of patients with nasopharyngeal cancer contained higher percentages of nucleic acids, collagen, phospholipids and phenylalanine but lower percentages of amino acids and carbohydrates. In addition, the researchers used principal component linear discriminant analysis (PCA-LDA) to identify a sensitivity and specificity of 90.7% and 100%, respectively, with the two types of plasma. However, serum SERS technology has yet to be used in diagnosing GB stones and GB polyps.

Multivariate statistical analysis combined with SERS for data analysis has been widely used in disease diagnosis [19]. Commonly used algorithms in multivariate statistical analysis include orthogonal partial least squares discriminant analysis (OPLS-DA), partial least squares discrimination analysis (PLS-DA), and principal component analysis (PCA). In contrast with PCA, OPLS-DA is a supervised statistical method used for DA, and its most important feature is that it can remove data variations in the independent variable X which are not related to those of categorical variable Y. Categorical information is mainly concentrated in a single principal component. Therefore, the model is simple and easy to interpret. In addition, its discriminant effect as well as visualization of the principal component score plot is more obvious [20]. OPLS-DA is used most commonly in metabolomics analysis [21,22]. Li et al., using SERS and OPLS-DA methods to study serum after total body irradiation in mice exposed to different radiation doses [23]. Kai et al., used SERS combined with OPLS-DA to differentiate between benign and malignant pleural effusions [24]. Driskell et al. studied the SERS spectra of eight rotavirus strains using PLS-DA and classified each strain at a >96% accuracy [25]. However, SERS combined with the OPLS-DA algorithm has yet to be applied to the classification diagnosis of GB stones and GB polyps.

In this study, we used SERS technology combined with multiple statistical analyses to establish a classification diagnostic model for healthy people and for patients with GB stones and GB polyps. The model obtained high diagnostic accuracy, revealing the tremendous potential that SERS technology has in differential diagnosis of benign GB diseases.

## 2. Materials and Methods

### 2.1. Serum Sample Collection and Preparation

In this case, 148 serum samples from the First Affiliated Hospital of Xinjiang Medical University were used in our investigation. Clinical diagnostics identified 51 cases of GB stone patients and 25 of GB polyp patients, with 72 cases identified as the healthy control group. Table 1 lists basic information about these participants, including their ages and gender. The medical ethics committee of the First Affiliated Hospital of Xinjiang Medical University approved the trial, and each patient completed an informed consent form (Approval No. K202107-16). Based on the standard operating protocols of clinical laboratories, blood samples were drawn. The serum was then separated from the blood samples by centrifuging them at 3000 rpm for 15 min. Finally, the serum samples were frozen (−80 °C) until the SERS measurement time was recorded.

### 2.2. Serum SERS Spectra Measurements

The SERS spectra of the serum samples were examined using a Raman micro-spectrometer (ATR6500-785, Optosky, China) coupled with a 785-nm laser [26]. The serum samples were defrosted at room temperature prior to SERS measurements. A 1:1 ratio of 5 µL each of serum and Ag colloid was then produced. A 10-µL sample was taken from this combination and placed on an aluminum slide. Following air-drying at room temperature, SERS spectra were recorded [27]. Ag colloid was purchased from Nanjing Jianzhi Instrument Equipment Co., Ltd., Nanjing, China. The preparation method of Ag nanoparticles (Ag NPs) was reported by Leopold and Lendl [28]. Briefly, 200 mL of 1.0 mM silver nitrate solution were first heated to a boil, and then 5.0 mL of 1% trisodium citrate were added dropwise with vigorous stirring. The mixture was then allowed to boil for an additional hour until it turned gray. Add distilled water to the solution after it has cooled so that the volume remains at 200 mL [14]. Prior to each online capture, a wavelength calibration was performed. The laser power and integration time were 5 mW and 3 s, respectively. The spectral range of the data was 600–1800 cm^−1^ and was collected using a 20× objective lens. Each sample was tested five times, and the average result was then used to determine the sample’s spectral data.

### 2.3. Spectral Data Pre-Processing

SERS data were preprocessed using smoothing, baseline correction and normalization methods. We used the SavitzkyGolay algorithm (order 5.9 points window) to smooth filter the collected serum spectral data, which not only improved the smoothness of the spectrum but also reduced the interference of noise [10]. The baseline was removed using the adaptive iteratively reweighted penalized least squares algorithm [29]. Vector normalization processing was conducted on each spectrum [30]. This processing was performed using MATLAB R2020a and Origin 64 software.

### 2.4. Data Analysis

We used SIMCA 14.0 software (Umetrics, Umea, Sweden) to conduct an analysis of Raman spectrum data. The principal component scores of the OPLS-DA models were used to accurately reflect the classification of diseases, and the performance of the OPLS model was assessed using the goodness-of-fit parameters R^2^ and Q^2^ [31] as related to the explained and predicted variances, respectively. The accurate performances of the diagnostic models under various illnesses were validated using a receiver operating characteristic (ROC) curve. The AUC values of these models were used to measure their quantitative performance, where higher AUC values indicated better model performance. The AUC values typically range from 0.5 to 1.0 [32]. Additionally, we used MATLAB R2021a software to perform PCA-LDA analysis and compared the results with the OPLA-DA classification model using three different metrics: sensitivity, specificity, and accuracy. As assessment criteria for the models, sensitivity and specificity were also used to gauge how well the models distinguished between patients with benign GB disease and controls. In this study, sensitivity refers to the probability that the model correctly diagnoses patients with benign GB disease [33]. In addition, the classifier and capacity of the OPLS-DA models in categorizing unidentified samples were assessed using 10-fold cross validation. For validation, the model was resampled 100 times under the null hypothesis using random permutations of the Y matrix [34].

## 3. Results and Discussion

### 3.1. Raman Spectral Analysis

Appendix A displays the spectral preprocessing of a representative serum sample from a healthy person. Due to instrument noise and outside ambient noise, denoising was first conducted to enhance the spectral signals in the spectrum collection operations of the serum samples. Note that the spectrum then had to be adjusted for baseline because the Raman signal was accompanied by the development of an autofluorescence signal in the biological material. Finally, each spectrum underwent vector normalization.

In this investigation, the Ag colloid served as an improved substrate. The results of an Ag nanoparticle (NP) UV absorption spectra and transmission electron microscopy micrograph with a 50-nm bar are shown in Appendix A. The UV absorption spectra at a high absorption peak of 417 nm demonstrated the purity of these Ag NPs [35]. Ag NPs used in this measurement had excellent purity. We obtained the SERS and Raman spectra from the same GB-stone serum patient to demonstrate the boosting effects of the Ag colloid. SERS signals were evident, as shown in Figure 1, demonstrating that our Ag colloid considerably enhanced the serum’s Raman spectra. Therefore, as a substrate, the Ag colloid may be more effectively used.

Datasets from the SRES spectra of 51 patients with GB stones, 25 patients with GB polyps and 72 healthy control subjects were gathered. A spectral data range of 600–1800 cm^−1^ was studied and provided the most diagnostically helpful information. Figure 2a–c shows the mean and difference spectrograms for GB stones and GB polyps, GB stones and healthy controls, and GB polyps and healthy controls, respectively, where the shaded areas represent the standard deviations of the means. The characteristic peaks of the three groups were mainly distributed in 637, 722, 810, 888, 1003, 1134, 1203, 1333, 1432, 1557 and 1652 cm^−1^. The histogram of the average intensity values of the serum SERS peaks with corresponding standard deviations is shown in Figure 2d. One-way analysis of variance was used to identify the significantly different peaks across the three groups with a *p* value (i.e., probability) cut-off of 0.05. These variations showed the changes in the components of serum biomolecules with GB disease progress. The attribution of each spectral peak is shown in Table 2 [13,14,15,17,36,37,38]. Based on a comparison of GB stones and GB polyps, the main peaks of difference characteristics were found to be 637 cm^−1^ (L-tyrosine, lactose), 722 cm^−1^ (coenzyme A), 1203 cm^−1^ (phenylalanine), 1432 cm^−1^ (D-glucosamine) and 1652 cm^−1^ (lipids), with all peaks showing statistical significance (*p* < 0.05). The characteristic peaks of GB stones and healthy controls were mainly distributed in 637 cm^−1^ (L-tyrosine, lactose), 722 cm^−1^ (coenzyme A), 1203 cm^−1^ (phenylalanine) and 1652 cm^−1^ (lipids), with all peaks showing statistical significance (*p* < 0.05). The characteristic peaks of GB polyps and healthy controls were mainly distributed in 637 cm^−1^ (L-tyrosine, lactose), 722 cm^−1^ (coenzyme A), 1134 cm^−1^ (D-mannose), 1203 cm^−1^ (phenylalanine), 1432 cm^−1^ (D-glucosamine) and 1652 cm^−1^ (lipids), with all peaks showing statistical significance (*p* < 0.05). At 637 cm^−1^ (L-tyrosine, lactose) and 1134 cm^−1^ (D-mannose) characteristic peaks, a significant upregulation or downregulation between the GB stone and GB polyp groups was observed, indicating the presence of abnormalities in serum glucose metabolism. Previous studies have reported that GB polyps have precancerous potential [39] and therefore are associated with glucose metabolism, which has been reported in other cancer research [18]. At 722 cm^−1^ (coenzyme A), the three groups exhibited significant differences, where coenzyme was a major factor in regulating sugar and fat as well as protein metabolism. In addition, at 1652 cm^−1^ (lipids), significant differences were observed between the three groups. The occurrence of GB polyps is generally believed to be closely related to cholesterol metabolism, and abnormal lipid metabolism may promote the formation of GB polyps [40]. The serum SERS signal of GB stone patients was significantly lower than that of healthy individuals at 888 cm^−1^ (Glutathione) peaks, indicating a decrease in the percentage of amino acids in the serum of GB stone patients, similar phenomena have been found in other areas of cancer research [41]. The serum SERS signal in the GB polyp group was higher than that in the healthy group at 1203 cm^−1^ (phenylalanine). It can be seen that the content of Phenylalanine in the serum of GB patients was significantly higher than that of healthy people, this may be because GB polyp was the cause of precancerous lesions, which was present in cervical cancer and other cancers [42]. Some studies have reported that due to the specific anatomical location of the gallbladder, when damage to the gallbladder occurs, obstruction of bile flow can affect the metabolic function of the liver, resulting in disorders of lipid metabolism and amino acid metabolism [3,43]. In our study, the differences between the spectral characteristic peaks suggested that the differential markers of the three groups were likely related to lipid or amino acid metabolism. Although differences were observed in certain SERS characteristic peaks of the three groups of serum, the three groups had similar spectral profiles. As a powerful classification algorithm, OPLS-DA and PCA-LDA was used to develop the classification model, to provide accurate results in measuring SERS performance, and thus to distinguish among the serum of GB stone and GB polyp patients and that of healthy controls.

### 3.2. Multivariate Analysis and Classification

The SERS spectral differences of the three types of serum were investigated using the OPLS-DA multivariate algorithm to evaluate the accuracy of screening utilizing serum SERS spectra. To create the model, 148 samples were used. We first compared the differences in serum SERS spectra of GB stones, GB polyps, and healthy controls using the supervised learning OPLS-DA of the SIMCA software, and we modelled the three types of samples for comparison. The score plot of the three group comparisons is shown in Figure 3a, where each point on the plot represents a sample. The horizontal and vertical coordinates represent the score values of principal components 1 and 2, respectively, and the ellipse represents the 95% confidence interval of the full sample analytical results. The plot shows significant differences among the serum SERS spectra of GB stones, GB polyps, and healthy individuals. The method can thus be used to clearly distinguish the three groups, where a clear tendency of separation is revealed between groups and aggregation within groups. Based on the following criteria, the robustness of these models was evaluated. R^2^X (cum), R^2^Y (cum) and Q^2^ (cum) are cumulative sums of squares (SS) of all x (PC) and y (SS) variables explained by all extracted components. Q^2^ (cum) is the proportion of all x (PC) and y variables that can be predicted for the extracted component [44]. For the triple classification model, R^2^X (cum) = 0.89, R^2^Y (cum) = 0.714 and Q^2^ (cum) = 0.609, indicating that the model has good predictive power. The permutation test was used to check the validity of the model and overfitting of the algorithm [34]. SIMCA was used to conduct 100 permutation tests on the dataset. The results of the three classification models are shown in Figure 4a, where the R^2^ and Q^2^ intercepts were 0.16 and −0.388, respectively. In general, when R^2^ was less than 0.3–0.4 and Q^2^ was less than 0.05, the model could be considered well-constructed. The results of this analysis demonstrated that the model developed by the OPLS-DA method was stable and could be applied to the classification of the three types of serum SERS spectra.

We also established a binary classification model based on OPLS-DA. Figure 3b–e shows the score plots of GB stones and GB polyps, GB stones and the healthy group, GB polyps and the healthy group, the case group (GB stones and GB polyps) and healthy group, respectively. Good quality parameters were shown by the OPLS-DA model, where R^2^X (cum), R^2^Y (cum) and Q^2^ (cum) for the classification models of GB stones and GB polyps were 0.82, 0.76 and 0.63, respectively. For the GB stones and healthy group, these same parameters were 0.84, 0.76 and 0.68, respectively. For the GB polyps and healthy group, they were 0.82, 0.81 and 0.64, respectively. Finally, for the case and healthy groups, they were 0.85, 0.76 and 0.69, respectively. Figure 4b–e shows the results of 100 permutation tests, where the intercepts of R^2^ and Q^2^ were 0.282, 0.172, 0.245, 0.171 and −0.562, −0.301, −0.462, −0.40, respectively. The intercepts of R^2^ and Q^2^ were less than 0.30 and 0.05, respectively, indicating that the model showed good robustness. This indicated that no overfitting occurs in our model, and the model has good prediction ability. These findings further reveal the effectiveness of the OPLS-DA-based serum SERS spectral classification method in differentiating between the two types of serum samples.

In addition, we used the PCA-LDA algorithm to classify and diagnose three sets of SERS data in order to compare with each other with the OPLS-DA algorithm. First, PCA was performed to reduce the dimensionality of the spectral dataset and extract PC features. The score plot of the three group comparisons was shown in Appendix A, and we can see the classification effect of the PCA-LDA algorithm on the three sets of serum. The loading plot of the first PC (PC1), which was responsible for 41.4% of the overall variance, was displayed in Appendix A. The findings of PC1 loading can be found to be in good accord with the variations in SERS spectra between the groups depicted in Figure 2a. Appendix A shows the PCA score plots for groups GB stones and GB polyps, GB stones and the healthy group, GB polyps and the healthy group, the case group and healthy group, respectively. It can be found that the PCA-LDA algorithm was significantly worse than the OPLS-DA algorithm in classifying the two groups of serum.

The confusion matrix of the triple classification results based on the OPLS-DA and PCA-LDA algorithm was shown in Table 3. The overall classification accuracy of the OPLS-DA and PCA-LDA algorithms were 92.6% and 83.8%, and the diagnostic sensitivity for the healthy, GB stone, and GB polyp groups were 98.60%, 90.20%, 80.00% and 83.30%, 84.30%, 84.00%, respectively. Figure 5 shows the calculated ROC curves using SIMCA software, where AUC (healthy Group) = 0.993, AUC (GB stone) = 0.989 and AUC (GB polyp) = 0.987, and the AUCs of all three approximated 1, indicating that the OPLS-DA model exhibited a good classification effect. Our binary classification results were shown in Table 4, where the overall classification accuracy of the OPLS-DA algorithms was greater than 93%, and the AUC values were higher than 0.99 in all four groups. The overall classification accuracy of the PCA-LDA algorithm ranged from 80–90%, and the AUC values for the four groups ranged from 0.874–0.905. The table thus shows that the classification model based on SERS and combined with the OPLS-DA algorithm has good diagnostic efficacy for benign GB diseases. In a previous study, Tung et al. [45] used bile juices as test specimens and employed the SERS technique to identify GB stone and GB polyp patients. However, the model exhibited poor discriminatory ability. In addition, Jin et al. [46] attempted infrared spectroscopic identification of GB polyps and GB stones using bile juices as specimens, achieving an overall classification accuracy of 78.6%. In our study, we used serum as specimens and the SERS technique to classify and diagnose GB stones and GB polyps with high accuracy. The serum test we employed was a non-invasive test as compared with that using bile juices. Here, accurate classification and diagnosis of disease could be achieved with just a single drop of blood serum from a patient. SERS-based technologies are quick, reliable, and accurate in terms of disease diagnostics and molecular identification. In addition, label-free SERS can utilize extensive fingerprint data for diagnosis and screening without having to label participants [47]. This study is a preliminary exploration of serum SERS technology combined with machine learning algorithms for diagnosing benign GB disease. A future work will investigate the metabolomics and other studies of serum from patients with GB disease to identify their specific diagnostic markers.

## 4. Conclusions

This study demonstrated for the first time the feasibility of SERS in discriminating between GB stones and GB polyps using serum samples. A portable Raman spectrometer was employed to obtain the SERS spectra of GB stone and GB polyp patients and healthy controls with only a small amount of serum, and the changes in biochemical components in the serum of patients as compared with normal subjects was reflected in the differences among the spectra. The OPLS-DA and PCA-LDA algorithms were combined with SERS to classify the SERS spectra of different serum types. The results showed that the sensitivity and specificity of the OPLS-DA algorithm for classifying GB stones, GB polyps and healthy groups were better than those of the PCA-LDA algorithm. This preliminary study is expected to set up a new path in developing a new clinical method for detecting GB stones and GB polyps. Our next step will be to collect a greater number of serum samples and conduct a more detailed investigation to evaluate the feasibility of the method.

## Figures and Tables

**Figure 1 diagnostics-13-00619-f001:**
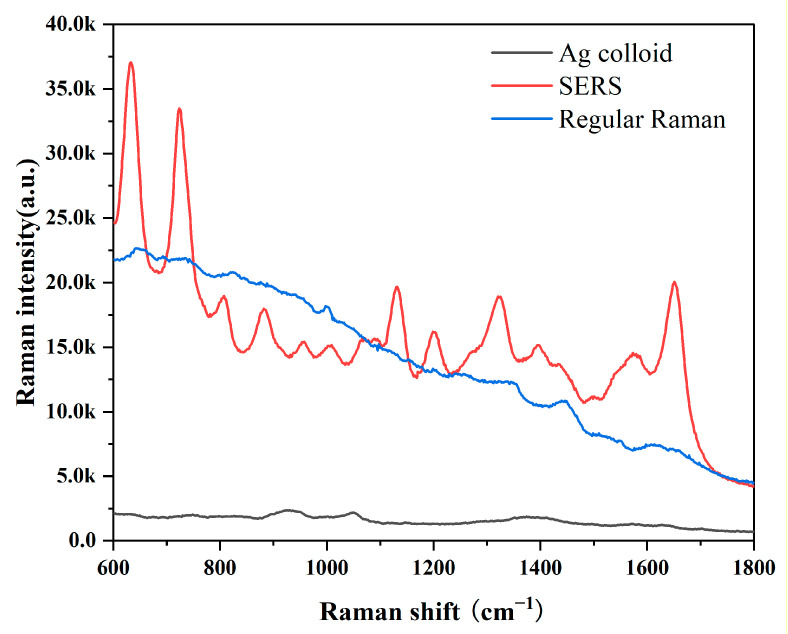
Comparison of SERS, RS and background Raman signal of Ag colloid; The blood serum patient with GB stone was mixed with Ag colloid in a 1:1 ratio to obtain SERS spectra of the serum (Red line); Conventional Raman spectra of free Ag colloid in the same sample (Blue line); Background Raman signal of Ag colloid (Black line).

**Figure 2 diagnostics-13-00619-f002:**
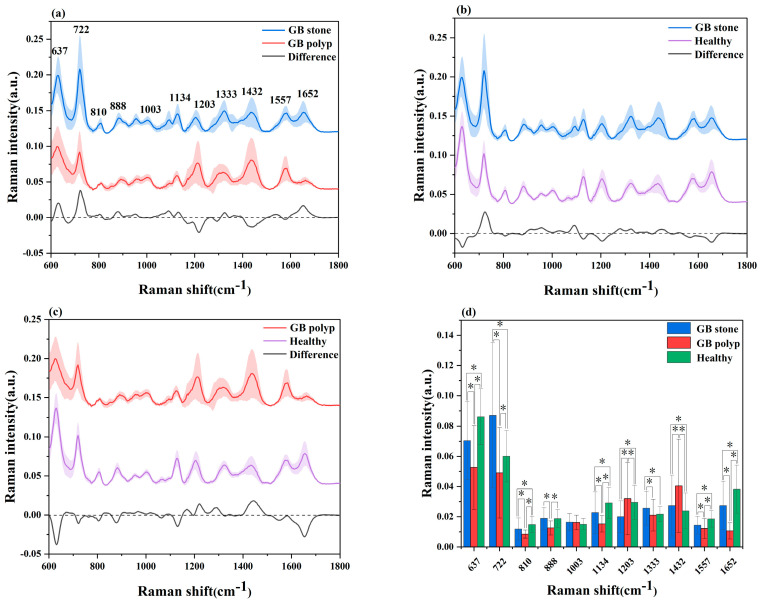
(**a**) Comparison of average SERS spectra of a GB stone and GB polyp patients, (**b**) GB stone and Healthy objects, (**c**) GB polyp and Healthy objects. The shaded areas represent the standard deviations of the means. In addition shown at the bottom is the difference spectrum. (**d**) The corresponding histograms of the average intensities and standard deviations of SERS peaks among the three groups. * *p* < 0.05.

**Figure 3 diagnostics-13-00619-f003:**
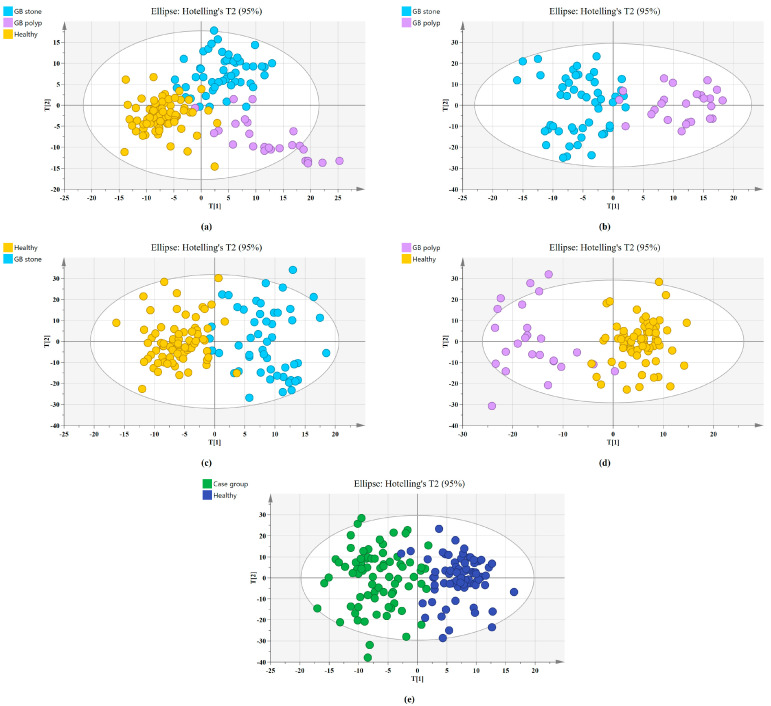
Principal component score plots of the SERS spectra based on OPLS-DA. (**a**) Score plot for Healthy, GB stone, and GB polyp serum samples. (**b**) Score plot for GB stone, and GB polyp serum samples. (**c**) Score plot for Healthy, and GB stone serum samples. (**d**) Score plot for Healthy, and GB polyp serum samples. (**e**) Score plot for Healthy, and Case group* serum samples. (*Case group, GB stone, and GB polyp Group).

**Figure 4 diagnostics-13-00619-f004:**
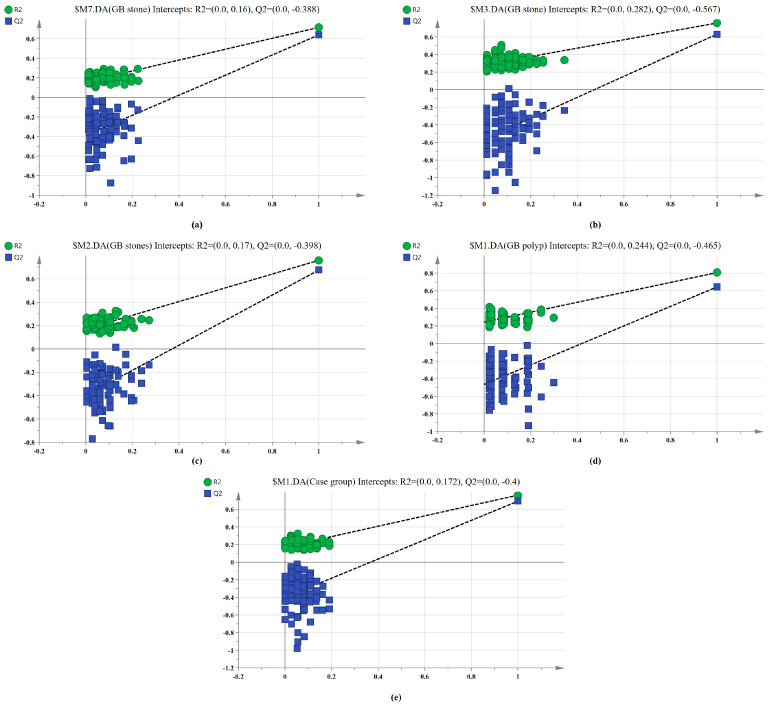
100-permutation test of the SERS data for three types of serum based on OPLS-DA. (**a**) Healthy, GB stone, and GB polyp classification models. (**b**) GB stone vs. GB polyp classification models. (**c**) Healthy vs. GB stone classification models. (**d**) Healthy vs. GB polyp classification models. (**e**) Healthy vs. Case group classification models.

**Figure 5 diagnostics-13-00619-f005:**
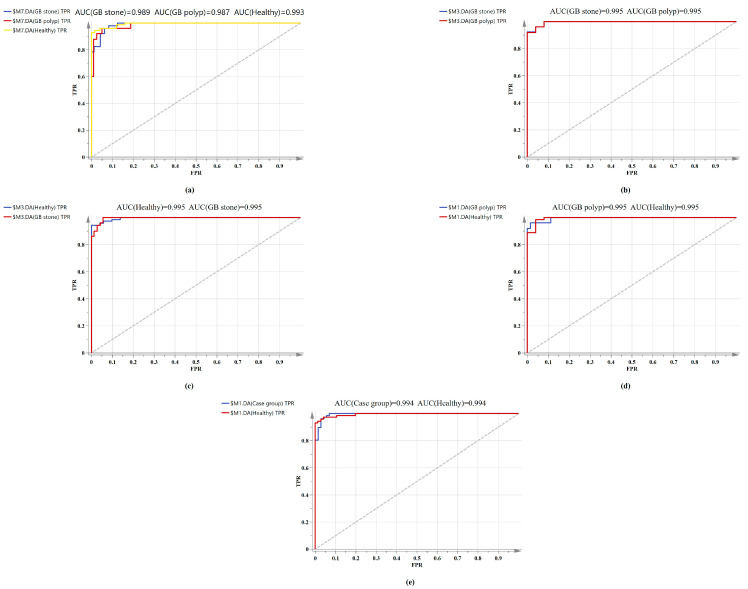
Comparison of ROC curves of serum SERS spectral utilizing the OPLS-DA algorithm. (**a**) Healthy vs. GB stone vs. and GB polyp serum samples. (**b**) GB stone serum samples vs. GB polyp serum samples. (**c**) Healthy serum samples vs. GB stone serum samples. (**d**) Healthy serum samples vs. GB polyp serum samples. (**e**) Healthy serum samples vs. Case group serum samples.

**Table 1 diagnostics-13-00619-t001:** Age and gender information of patients with GB stone, GB polyp, and healthy volunteers.

	GB Stone	GB Polyp	Healthy
**Gender**			
Male	27	11	42
Female	24	14	30
**Age**			
Mean	46.5	43	45.7
Median	44	42	41
Range	32–65	35–59	34–52

**Table 2 diagnostics-13-00619-t002:** Tentative assignments of main peaks observed in SERS spectra of serum samples according to the literature.

Raman Shift (cm^−1^)	Major Assignments
637	L-tyrosine, lactose
722	Coenzyme A
810	L-serine, glutathione
888	Tryptophan, glutathione
1003	Phenylalanine
1134	D-mannose
1203	L-tryptophan, phenylalanine
1333	Guanine, adenine
1432	D-glucosamine
1557	Tryptophan
1652	Lipids

**Table 3 diagnostics-13-00619-t003:** The results of triple classification confusion matrix based on OPLS-DA and PCA-LDA analysis.

Group	OPLS-DA			PCA-LDA		
	Healthy	GB Stone	GB Polyp	Healthy	GB Stone	GB Polyp
Healthy	71	1	0	60	3	9
GB stone	0	46	5	4	43	4
GB polyp	3	2	20	1	3	21
Sensitivity (%)	98.6	90.2	80	83.3	84.3	84
Specificity (%)	86.8	93.8	95.1	84.2	83.5	83.7
Accuracy (%)	92.6			83.8		
AUC	0.993	0.989	0.987	0.842	0.839	0.838

**Table 4 diagnostics-13-00619-t004:** The results of binary classification confusion matrix based on OPLS-DA and PCA-LDA analysis.

Group	OPLS-DA				PCA-LDA			
	Sensitivity (%)	Specificity (%)	Accuracy (%)	AUC	Sensitivity (%)	Specificity (%)	Accuracy (%)	AUC
GB stone GB polyp	98	88	94.7	0.995	86.3	88	86.8	0.874
GB stone	90.2	97.2	94.3	0.995	86.3	97.2	92.6	0.917
Healthy								
GB polyp	92	100	95.9	0.995	92	86.1	87.6	0.891
Healthy								
Case group *	90.8	97.2	93.9	0.994	88.2	93.1	90.5	0.905
Healthy								

* Case group, GB stone and GB polyp.

## Data Availability

Data underlying the results presented in this paper are not publicly available at this time but may be obtained from the authors upon reasonable request.

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
