# Peer review of "Rapid Identification of Benign Gallbladder Diseases Using Serum Surface-Enhanced Raman Spectroscopy Combined with Multivariate Statistical Analysis"

_diagnostics, 2023, doi:10.3390/diagnostics13040619_

Round 1

Reviewer 1 Report

In this paper, Dawuti and colleagues demonstrated a serum analysis to differentiate between patients with gallbladder stones, gallbladder polyps and healthy patients. It is very interesting that these different entities are reflected in the serum of patients/individuals. However, I have several issues.

- the included patients/individuals have either gallbladder stones, gallbladder polyps or are healthy. How is this objectified? By ultrasound? Why is a additional method (as presented in this paper) necessary?

- from a clinical point of view, differentiating between these groups is not really interesting. Patients with abdominal complaints related to GB stones will undergo a cholecystectomy, as will patients with a certain size of polyp. However, the most difficult diagnosis is differentiating between benign (polyp) and malignant disease (cancer). This difference is sometimes hard to make and therefore including these patients in the analysis would be of great interest. 

- a smaller not that in the Introduction 'tones' are written in stead of 'stones'

Author Response

Point 1: The included patients/individuals have either gallbladder stones, gallbladder polyps or are healthy. How is this objectified? By ultrasound? Why is a additional method (as presented in this paper) necessary?

Response 1: Thanks for your comments. Patients with gallbladder (GB) stones and GB polyps included in this study were hospitalized in the Department of Hepatobiliary Surgery of the First Affiliated Hospital of Xinjiang Medical University and were diagnosed using ultrasound. The healthy individuals were from health examination centers and all were diagnosed by ultrasound to rule out gallbladder disease.

According to in this paper, At present, abdominal B ultrasound, computed tomography (CT) and other imaging technologies are commonly used to identify GB stones and GB polyps, but they all require expensive software and hardware as well as visual observation of imaging physicians to determine results, they also have the problem of low sensitivity, and in community hospitals, rural hospitals and other small hospitals are not equipped with some large diagnostic equipment and professional physicians. In addition laboratory results have revealed the leukocytosis with a left shift and minimal increase in the levels of bilirubin and alka-line phosphatase. Overall, GB stones and GB polyps can be neither confirmed nor ruled out by a single clinical finding or laboratory test . Therefore, finding a rapid and accurate diagnostic method to identify benign GB disease is necessary.

Point 2: From a clinical point of view, differentiating between these groups is not really interesting. Patients with abdominal complaints related to GB stones will undergo a cholecystectomy, as will patients with a certain size of polyp. However, the most difficult diagnosis is differentiating between benign (polyp) and malignant disease (cancer). This difference is sometimes hard to make and therefore including these patients in the analysis would be of great interest.

Response 2: Thank you for your helpful advice. It is very important to distinguish benign and malignant gallbladder lesions in clinical practice. In addition, GB tones and GB polyps are the two most prevalent benign disorders and has a 5–10% probability of becoming cancerous (malignant). It is therefore crucial to classify benign gallbladder diseases in advance and to provide corresponding interventions. This study is a preliminary exploration of serum SERS technology combined with machine learning algorithms for the diagnosis of benign gallbladder disease. Since we currently have no serum samples related to malignant gallbladder diseases, we will spend more time and effort in the next study to apply for ethical and informed consent, and collect more serum samples from patients diagnosed with benign and malignant gallbladder diseases for further study.

Point 3: - a smaller not that in the Introduction 'tones' are written in stead of 'stones'

Response 3: Thanks for your comments. We apologize for some minor errors in the manuscript, and we have corrected the language and grammatical errors in the manuscript.

We have tried our best to revise our manuscript according to the comments. We would like to express our great appreciation to the editor and reviewers for comments on our paper and hope that these changes lead to the manuscript's acceptance.

Once again, thank you very much for your comments and suggestions.

We look forward to hearing from you.

Best wishes to you!

Sincerely yours,

Dr. Guodong Lü

Reviewer 2 Report

In this work, surface-enhanced Raman spectroscopy (SERS) was used to measure the blood samples from gallbladder (GB) stones and GB polyps. Through the spectral analysis and multivariate statistical analysis, three groups including GB stone, GB polyp and healthy subjects can be well classified. Some issues should be revised or clarified before researching the final recommendation.

1. The language and writing should be greatly improved by a native-English speaking person. Such as “OPLS-DA is used in metabolomics analysis The application is more frequent [21,22].”

2. Some abbreviations should be well defined when they firstly appear in the main text.

3. For Part 3.1 Raman spectral analysis, more discussions regarding the changes in the bio-components instead of simple description of increase or decrease should be added. Especially, the relation between the changes and disease formation or progress should be discussed.

4.  In this work, OPLS-DA was used to analyze the SERS data. I would like to see the comparison between the results by the OPLS-DA and PCA-LDA.

Author Response

Point 1: The language and writing should be greatly improved by a native-English speaking person. Such as “OPLS-DA is used in metabolomics analysis The application is more frequent [21,22].”

Response 1: Thanks for your comments. As per the reviewer's request, Language and writing have been improved by editage publish and flourish company (https://www.editage.cn/), and the relevant proofs have been uploaded to the journal submission website along with the response letter.

Point 2: Some abbreviations should be well defined when they firstly appear in the main text.

Response 2: Thank you for your helpful advice. According to the Reviewer’s suggestion, we have defined the abbreviations that appear for the first time in the text. The relevant revised contents were marked red in the manuscript.

Point 3: For Part 3.1 Raman spectral analysis, more discussions regarding the changes in the bio-components instead of simple description of increase or decrease should be added. Especially, the relation between the changes and disease formation or progress should be discussed.

Response 3: Thanks for your comments. According to the Reviewer’s suggestion, we have added more discussion on the changes in biological composition in section 3.1 Raman spectral analysis. The relevant revised contents were marked red in the manuscript.

The detail was listed below:

The occurrence of GB polyps is generally believed to be closely related to cholesterol me-tabolism, and abnormal lipid metabolism may promote the formation of GB polyps [39]. The serum SERS signal of GB stone patients was significantly lower than that of healthy individuals at 888 cm-1 (Glutathione) peaks, indicating a decrease in the percentage of amino acids in the serum of GB stone patients, similar phenomena have been found in other areas of cancer research [40]. The serum SERS signal in the GB polyp group was higher than that in the healthy group at 1203cm-1 (phenylalanine). It can be seen that the content of Phenylalanine in the serum of GB patients was significantly higher than that of healthy people, this may be because GB polyp was the cause of precancerous lesions, which was present in cervical cancer and other cancers [41]. Some studies have reported that due to the specific anatomical location of the gallbladder, when damage to the gallbladder occurs, obstruction of bile flow can affect the metabolic function of the liver, resulting in disorders of lipid metabolism and amino acid metabolism [3][42].

References:

[40] S. Feng, R. Chen, J. Lin, J. Pan, Y. Wu, Y. Li, J. Chen, H. Zeng, Gastric cancer detection based on blood plasma surface-enhanced Raman spectroscopy excited by polarized laser light. Biosens Bioelectron. 2011, 26, 3167-3174.

[41] S. Feng, D. Lin, J. Lin, B. Li, Z. Huang, G. Chen, W. Zhang, L. Wang, J. Pan, R. Chen, H. Zeng, Blood plasma surface-enhanced Raman spectroscopy for non-invasive optical detection of cervical cancer. Analyst. 2013,138, 3967-3974.

[42] M. van Dooren, P.R. de Reuver, Gallbladder polyps and the challenge of distinguishing benign lesions from cancer. United European Gastroenterol J. 2022,10, 625-626.

Point 4: In this work, OPLS-DA was used to analyze the SERS data. I would like to see the comparison between the results by the OPLS-DA and PCA-LDA.

Response 4: Thank you for your helpful advice. According to the Reviewer’s suggestion, The classification results of PCA-LDA algorithm were added to the text, and compared with the results of OPLS-DA algorithm. The relevant revised contents were marked red in the manuscript.

The detail was listed below:

In addition, we used the PCA-LDA algorithm to classify and diagnose three sets of SERS data in order to compare with each other with the OPLS-DA algorithm. First, PCA was performed to reduce the dimensionality of the spectral data set and extract PC fea-tures. The score plot of the three group comparisons was shown in Figure S3(a), and we can see the classification effect of the PCA-LDA algorithm on the three sets of serum. The loading plot of the first PC (PC1), which was responsible for 41.4% of the overall variance, was displayed in Figure S3(b). The findings of PC1 loading can be found to be in good ac-cord with the variations in SERS spectra between the groups depicted in Figure 2(a). Figure S4(a)–(d) shows the PCA score plots for groups GB stones and GB polyps, GB stones and the healthy group, GB polyps and the healthy group, the case group and healthy group, respectively. It can be found that the PCA-LDA algorithm was significantly worse than the OPLS-DA algorithm in classifying the two groups of serum.

The confusion matrix of the triple classification results based on the OPLS-DA and PCA-LDA algorithm was shown in Table 3. The overall classification accuracy of the OPLS-DA and PCA-LDA algorithms were 92.6% and 83.8%, and the diagnostic sensitivity for the healthy, GB stone, and GB polyp groups were 98.60%, 90.20%, 80.00% and 83.30%, 84.30%,84.00%, respectively. Figure 5 shows the calculated ROC curves using SIMCA software, where AUC (healthy Group) = 0.993, AUC (GB stone) = 0.989 and AUC (GB polyp) = 0.987, and the AUCs of all three approximated 1, indicating that the OPLS-DA model exhibited a good classification effect. Our binary classification results were shown in Table 4, where the overall classification accuracy of the OPLS-DA algorithms was greater than 93%, and the AUC values were higher than 0.99 in all four groups. The overall classification accuracy of the PCA-LDA algorithm ranged from 80-90%, and the AUC values for the four groups ranged from 0.874-0.905.

Table 3. Results of triple classification confusion matrix based on OPLS-DA and PCA-LDA analysis.

Group

OPLS-DA

PCA-LDA

Healthy

GB stone

GB polyp

Healthy

GB stone

GB polyp

Healthy

71

1

0

60

3

9

GB stone

0

46

5

4

43

4

GB polyp

3

2

20

1

3

21

Sensitivity (%)

98.6

90.2

80

83.3

84.3

84

Specificity (%)

86.8

93.8

95.1

84.2

83.5

83.7

Accuracy (%)

92.6

83.8

AUC

0.993

0.989

0.987

0.842

0.839

0.838

Table 4. Results of binary classification confusion matrix based on OPLS-DA and PCA-LDA analysis.

Group

OPLS-DA

PCA-LDA

Sensitivity(%)

Specificity(%)

Accuracy(%)

AUC

Sensitivity(%)

Specificity(%)

Accuracy(%)

AUC

GB stone

GB polyp

98

88

94.7

0.995

86.3

88

86.8

0.874

GB stone

90.2

97.2

94.3

0.995

86.3

97.2

92.6

0.917

Healthy

GB polyp

92

100

95.9

0.995

92

86.1

87.6

0.891

Healthy

Case group*

90.8

97.2

93.9

0.994

88.2

93.1

90.5

0.905

Healthy

*Case group, GB stone and GB polyp.

Supplementary Materials:

Figure.S3. (a) Principal component score plots of the SERS spectra based on PCA-LDA; (b) Loading plot for the first PC. The first PC accounting for 41.4% of the total variance.

Figure.S4. Principal component score plots of the SERS spectra based on PCA-LDA. (a) Score plot for GB stone, and GB polyp serum samples. (b) Score plot for Healthy, and GB stone serum samples. (c) Score plot for Healthy, and GB polyp serum samples. (d) Score plot for Healthy, and Case group serum samples.

We have tried our best to revise our manuscript according to the comments. We would like to express our great appreciation to the editor and reviewers for comments on our paper and hope that these changes lead to the manuscript's acceptance.

Once again, thank you very much for your comments and suggestions.

We look forward to hearing from you.

Best wishes to you!

Sincerely yours,

Dr. Guodong Lü

Round 2

Reviewer 2 Report

The revised version now can be accepted for publication in this journal.

Author Response

A: Thanks for your comments. In accordance with the reviewers' comments, we checked all references of the article.

B: We describe the process of silver glue preparation for SERS.

The preparation method of Ag nanoparticles (Ag NPs) was reported by Leopold and Lendl [28]. Briefly, 200 ml of 1.0 mM silver nitrate solution were first heated to a boil, and then 5.0 ml of 1% trisodium citrate were added dropwise with vigorous stirring. The mix-ture was then allowed to boil for an additional hour until it turned gray. Add distilled water to the solution after it has cooled so that the volume remains at 200 ml [29].

References

  1. Leopold, B. Lendl, A new method for fast preparation of highly surfaceenhanced Raman scattering (SERS) active silver colloids at room temperature by reduction of silver nitrate with hydroxylamine hydrochloride. Phys Chem. B. 2003, 107 (24), 5723–5727.
  2. Kashif, M. I. Majeed, M. A. Hanif, and A. ur Rehman, Surface enhanced Raman spectroscopy of the serum samples for the diagnosis of Hepatitis C and prediction of the viral loads, Spectrochim. Acta, Part A. 2020, 242, 118729.

C:We have analyzed the noise of the surrounding environment.

The ambient noise here refers to the illumination, light, illumination of other instruments and some man-made noise in the laboratory environment when Raman spectra are measured.
